# The Influence of Mineral NPK Fertiliser Rates on Potassium Dynamics in Soil: Data from a Long-Term Agricultural Plant Fertilisation Experiment

**DOI:** 10.3390/plants12213700

**Published:** 2023-10-27

**Authors:** Jonas Arbačauskas, Zigmas Jonas Vaišvila, Gediminas Staugaitis, Lina Žičkienė, Aistė Masevičienė, Donatas Šumskis

**Affiliations:** Agrochemical Research Laboratory, Institute of Agriculture, Lithuanian Research Centre for Agriculture and Forestry, Savanoriu Av. 287, LT-50127 Kaunas, Lithuania; jonas.arbacauskas@lammc.lt (J.A.); zigmas@vaisvila.eu (Z.J.V.); gediminas.staugaitis@lammc.lt (G.S.); lina.zickiene@lammc.lt (L.Ž.); aiste.maseviciene@lammc.lt (A.M.)

**Keywords:** potassium fertilisers, nutrient interaction, metabolizable energy, balance, soil

## Abstract

A fertilisation experiment, with the aim to determine the effects of different potassium fertiliser rates and their interactions with nitrogen and phosphorus on field-rotation productivity, potassium balance, fertiliser utilization, and changes in the content of potassium in soil, was carried out in Lithuania between 1971 and 2020. The multi-factorial scheme with 45 treatment plots, where seven rates (including zero) of nitrogen, phosphorus, and potassium fertilisers were studied. The experimental treatments during the study period were carried out on winter wheat, spring wheat, spring barley, sugar beet, spring rapeseed, and annual and perennial grasses. It was found that potassium fertilisers were the most effective on agricultural crops when used in combination with other major plant nutrients—i.e., nitrogen and phosphorus. The required balance of potassium (K_2_O) in the soil was measured, when nitrogen and phosphorus fertilisers were applied together to compensate for potassium removal; when applying low nitrogen (N) (72 kg ha^−1^) and phosphorus (P_2_O_5_) (64 kg ha^−1^) fertiliser rates, 128 kg ha^−1^ of potassium fertilisers are required. When using high nitrogen (180 kg ha^−1^) and phosphorus (160 kg ha^−1^) fertiliser rates, 160 kg ha^−1^ of potassium is needed. The highest potassium uptake, reaching 51.6%, was achieved when plants had been fertilised with nitrogen (108 kg ha^−1^), phosphorus (96 kg ha^−1^), and potassium (96 kg ha^−1^). When fertilising with potassium fertilisers alone, the content of plant-available K_2_O content in the soil increased, whereas with fertilisation with nitrogen and phosphorus combined K_2_O content is decreased, except in the plots where the plants had been fertilised with potassium fertiliser at rates of 128 kg ha^−1^ and higher. Due to the influence of fertilisers, the amount of non-exchangeable potassium in the soil also increased, but relatively little compared to the amount of available potassium content. Thus, one of the main conditions for the effective use of potassium fertilisers is ensuring optimal plant nutrition with other nutrition elements, especially nitrogen and phosphorus.

## 1. Introduction

Potassium is one of the main plant nutrients, along with nitrogen and phosphorus. It influences many physiological processes affecting plant growth, yield, and quality, as well as plant resistance to pathogens, frost, heat, drought, pesticides, and heavy metals [1,2]. Moreover, potassium is almost as important as nitrogen for plant nutrition and its impact on crop yields in low-potassium soils is no less than that of nitrogen and phosphorus [3,4]. Potassium deficiency in agricultural crops reduces yield, quality, and stress tolerance [5,6]. This is particularly important in recent times, as droughts and heat stresses due to climate change are expected to have a negative impact on the yield of agricultural crops [7]. However, potassium fertiliser use is significantly lower worldwide compared to nitrogen and phosphorus, which imbalances the relationship between these nutrients in the soil and plants [2,8]. The interactions of nitrogen with potassium and/or phosphorus lead to improved root development and other plant functions that influence yield and quality [9,10]. Phosphorus and potassium fertilisers have a significant impact on the efficiency of nitrogen fertilisers, as well as on phosphorus and potassium dynamics in soil, leading to an increase in the efficiency of nitrogen and phosphorus fertiliser use [4,11,12]. Improved nitrogen uptake also reduces nitrate pollution [13]. On the other hand, the effectiveness of potassium fertilisers on agricultural crops also depends on their nutrition with other nutrients, especially nitrogen and phosphorus [12,14]. Furthermore, it has been noted that balanced application of nitrogen, phosphorus, and potassium fertilisers to agricultural crops results in a five to ten times more efficient uptake of fertilisers compared to applying nitrogen fertilisers alone [12].

Potassium reserves in the soil are high; however, not all of them are readily available to plants [7]. Most of the potassium, 90–98%, is found in soil minerals such as feldspar and mica, and only a small fraction of the potassium in this form is available to plants. Another source of soil potassium is non-exchangeable potassium, which accounts for up to 10% of the total potassium content. Non-exchangeable potassium is found in 2:1 clay minerals and is not taken up much by plants. The third source of soil potassium is potassium directly taken up by plants (1–2%), which is present in the exchangeable form and in soil solution [15,16]. Currently, due to the reduced use of potassium fertilisers, potassium uptake by plants is decreasing [17]. The main causes of soil potassium deficiency are the removal of potassium with crop yields, losses due to leaching and soil erosion, and inefficient use of potassium fertilisers [8,18,19].

The demand for mineral potassium fertilisers is increasing rapidly in developing and populous countries, as well as in some developed countries [7,20]. For example, a quarter of China’s agricultural land is low in potassium [2,18,19,21]. In India, on average, potassium inputs from fertilisers are as much as 3–5 times lower compared to its accumulation in the yield of agricultural crops, and the inputs of this nutrient for plant growth are about seven times less than nitrogen and three times less than phosphorus [22]. As noted by other sources [2,23], potassium is the main yield-limiting factor in as much as 72% of India’s agricultural land. The potassium balance in soil is also affected by unbalanced application of nitrogen, phosphorus, and potassium fertilisers, straw removal, as well as recent introduction of higher-yielding crop varieties [24,25].

In Lithuania, 16.8% of soils had low potassium content (<100 mg kg^−1^) between 1995 and 2006 [26]. However, in some countries, such as Germany and Switzerland, more potassium is incorporated into soil with fertiliser than is taken up by plants [27,28]. However, potassium fertiliser resources are limited and non-renewable worldwide, and need to be used efficiently [2,8].

Achieving stable and high crop yields, maintaining adequate nutrient levels in soil, and reducing potassium leaching require sufficient attention to the potassium balance in soil [17,29]. Fertilisation recommendations for agricultural crops are usually based on studies of the exchangeable potassium content in the topsoil, as the effectiveness of potassium fertilisers for agricultural crops depends on the content of this element in soil [2]. However, the effect of potassium fertilisers on agricultural crops depends not only on the exchangeable potassium but also on the concentration of non-exchangeable potassium in soil, as well as on soil texture, cation exchange capacity, soil moisture, the presence of other plant nutrients in soil, and the type of crop [15,22,30]. In addition, soil potassium content is influenced by soil-forming rock, weathering intensity, application of organic and mineral fertilisers, leaching, erosion, and potassium export with crop yield [8]. Potassium is most commonly deficient in acidic sandy soils, as well as in waterlogged and saline soils [15].

The effects of fertilisation on crop yields and soil nutrient balance are more objectively assessed using the results of long-term fertilisation experiments on agricultural crops, because it reduces the influence of weather conditions on the results of research [17,31,32].

It can be hypothesised that potassium fertilisers have a greater influence on the productivity of field crop rotation if they are used together with nitrogen and phosphorus fertilisers.

Therefore, the aim of this work was to determine the effect of long-term use of potassium fertilisers and their interaction with nitrogen and phosphorus on agricultural plants and soil.

## 2. Materials and Methods

### 2.1. Location and Climate Conditions

The experiment was carried out for 50 years (1971–2020) in central Lithuania at Skėmiai (55°34′29″ N and 23°45′1″ E), 95 m above sea level. The soil used is sandy loam Epicalcari-Endocalcari-Endohypogleyic Luvisol (WRB, 2015). Its physical and agrochemical characteristics are presented in (Table 1).

Most of Lithuania has a temperate (boreal) climate, with an average temperature in the coldest month of below −3 °C and an average temperature in the hottest month of below 22 °C. There is sufficient rainfall in all seasons, with snow cover in winter. The country has a warm season of 230–250 days and a growing season of 185–196 days [33]. In Lithuania, 32% of precipitation is converted into runoff, and the rest (68%) is returned to the atmosphere through evaporation processes [34].

Between 1971 and 2020, the average annual air temperature in the experimental area located in Central Lithuania was 7.1 °C, and the average annual precipitation was 572 mm.

### 2.2. Experimental Design and Details

The research was carried out according to a 45-treatment multi-factor design (Π3) developed by V. Peregudov, in which seven rates of nitrogen (including zero), phosphorus, and potassium fertilisers were investigated [35]. The design consisted of three designs: a 27-treatment 3 × 3 × 3 design, in which three nitrogen, phosphorus, and potassium fertiliser rates (0, 3, and 6) were studied; an 8-treatment design of 2 × 2 × 2, with two nitrogen, phosphorus, and potassium fertiliser rates (1 and 5), supplemented by the central design treatment 333 (9 treatments in total); and an 8-treatment design of 2 × 2 × 2, with two nitrogen, phosphorus, and potassium fertiliser rates (2 and 4), supplemented by the central design treatment 333 (9 treatments in total). Here, the first place value indicates nitrogen, the second indicates phosphorus, and the third indicates potassium fertiliser rates, where on average over the experimental period, the rates are N = 36 kg ha^−1^, P_2_O_5_ = 32 kg ha^−1^, and K_2_O = 32 kg ha^−1^. The treatments were conducted with two replicates. The specific NPK rates are given in the results tables, where the abbreviations used are: N_0_, N_108_, and N_216_—0, 108, and 216 kg ha^−1^ of nitrogen (N) applied annually, respectively; P_0_, P_96_, and P_192_—0, 96, and 192 kg ha^−1^ of phosphorus (P_2_O_5_) applied annually, respectively; and K_0_, K_96_, and K_192_—0, 96, and 192 kg ha^−1^ of potassium (K_2_O) applied annually, respectively. The total size of each experimental plot was 6 × 9 m. The fertilisers used were ammonium nitrate, granular superphosphate, and potassium chloride crystals.

During different periods of the experiment, the following crops were grown: 10 years of winter wheat; 2 years of spring wheat; 10 years of spring barley; 6 years of sugar beet; 4 years of spring oilseed rape; 10 years of annual forage legumes; and 8 years of perennial grasses.

### 2.3. Sampling and Methods of Chemical Analysis

Soil samples for general soil characterisation of the experimental area were taken prior to the setting up of the experiment in spring 1971, and for the determination of available potassium (K_2_O) from each of 90 fields in autumn 2020, after the spring barley harvest. Samples were taken from the 0–20 cm soil layer with a 1.5 cm diameter drill, with 15 punctures per pooled sample per field. Soil analyses were carried out on air-dried samples after screening the soil through a 2 mm sieve.

The concentration of available potassium (K_2_O) in soil was determined by the Egner–Riehm–Domingo (A–L) method [36]. Soil available K_2_O was extracted using a 1:20 (wt vol^−1^) soil suspension of ammonium lactate–acetic acid extractant (pH 3.7). The suspension was shaken for 4 h. Mobile K_2_O was determined using flame emission spectroscopy with a flame photometer Scherwood M410.

Soil-exchangeable and non-exchangeable potassium content in soil were determined in 54 fields using the 1 mol L^−1^ hot nitric acid (HNO_3_) extraction method [37]. The finely ground soil was gently heated with nitric acid (HNO_3_) in an Erlenmeyer flask for 15 min from the start of boiling. The cooled sample was filtered, and the extract was diluted to 100 mL with 0.1 mol L^−1^ HNO_3_. The K concentration of the extract solution was determined by an atomic absorption spectrometer—Analyst 200. The amount of non-exchangeable potassium in the soil was calculated from the amount of potassium dissolved in the 0.1 mol L^−1^ HNO_3_ minus the amount of this element obtained by the A–L method.

According to the soil bulk density, and potassium content in the soil, the resulting mg kg^−1^ was converted to kg ha^−1^.

The determination of soil pH was performed using a 1:5 (vol vol^−1^) soil suspension in 1 M KCl.

Organic carbon (C_org_), according to ISO 10694:1995 [38], underwent dry combustion with total carbon analyser Liqui TOC II.

Potassium (K) samples from plants were burned in a muffle for 6–8 h at a temperature of 550 degrees Celsius. Next, after adding nitric acid, it was evaporated to dryness for 1 h. It was further treated with 20% HCl, heated again to dryness and washed with distilled water. Potassium content was determined using a flame photometer—JENWAY PFP7.

### 2.4. Calculations and Data Analysis

The average yield of agricultural crops’ main and secondary productions over the entire 50-year study period was calculated in terms of the energy value of the yield, using the average values of data from analyses carried out in Lithuania for individual crops [39].

The potassium balance in soil (B) is calculated using the formula:B kg ha^−1^ = T − D,
where T is the amount of potassium (K_2_O) applied to the soil with fertiliser in kg ha^−1^; D is the amount of potassium (K_2_O) accumulated in the crop and removed from the field in kg ha^−1^.

The harvest of the main and secondary crops has been removed from the field. Potassium losses due to leaching were not included in the balance sheet calculation.

The uptake of potassium from mineral fertilisers (UK%) was calculated using the formula:$$UK\%=\frac{K-N}{F}\times 100,$$
where F is the potassium (K_2_O) fertiliser rate (kg ha^−1^); and K and N are the amounts of potassium (K_2_O) accumulated in the plants (kg ha^−1^) in potassium-fertilised and in unfertilised plots, respectively.

Statistical significance of the experimental data was assessed using Duncan’s Multiple Range test; significant differences were established between the data lettered a, b, c, d, e, f, etc., at 5% probability level (*p* ≤ 0.05). Means and their ratios as well as standard deviations (SD) were calculated using the software Microsoft Office Excel 2010. To determine the strength and nature of the relationship between the variables, correlation and regression data analysis was performed using the software STATISTICA 7 [40,41].

## 3. Results

### 3.1. Field Crop Productivity

In order to compare the productivities of plants of different harvest values, they were evaluated by their metabolic energy. The effectiveness of potassium fertiliser on the productivity of field rotation significantly depended not only on the rate of potassium fertiliser, but also on the nitrogen and phosphorus nutrition of the crop. This can be seen in Table 2; applying the average annual fertilisation rates of 96 and 192 kg ha^−1^ K_2_O, but without nitrogen and phosphorus fertiliser, did not lead to a statistically significant increase in the amount of metabolisable energy when comparing the above fertiliser rates with each other. However, there were significant increases of 16.3 and 17.4%, respectively, in the metabolisable energy of those rates, compared with the 40.3 GJ ha^−1^ obtained in the unfertilised plots. In contrast, fertilisation with 96 and 192 kg ha^−1^ K_2_O, in combination with nitrogen and phosphorus fertilisers (N_108_P_96_), resulted in a significant increase in the amount of metabolisable energy of up to 98.7% and 107.9%, respectively, compared with the unfertilised plots.

Fertilisation with potassium and phosphorus fertilisers significantly increased the efficiency of nitrogen fertiliser. Fertilising agricultural crops with 216 kg ha^−1^ of nitrogen, but not with phosphorus and potassium, resulted in an average annual production of 57.0 GJ ha^−1^ of metabolisable energy over the study period. Fertilisation with N_216_P_0_K_96_ showed a 12.4% increase in metabolisable energy when compared to plots fertilised with nitrogen fertiliser (N_216_P_0_K_0_) alone. However, the highest increase in metabolisable energy, of 53.2%, over the study period was obtained when agricultural crops were fertilised annually with the maximum fertiliser rate (N_216_P_192_K_192_) compared to those fertilised with nitrogen only at a rate of 216 kg ha^−1^ but without phosphorus and potassium fertilisers. The studies show that there was no reliable reduction in metabolisable energy when agricultural crops were also fertilised annually with lower fertiliser rates of 108 kg ha^−1^ of nitrogen, 96 kg ha^−1^ of phosphorus (P_2_O_5_), and 96 kg ha^−1^ of potassium (K_2_O).

Statistical correlation analysis of the results showed a very strong correlation (R = 0.94; *p* ≤ 0.05) between the metabolisable energy content in GJ ha^−1^ and NPK fertiliser rates and their interactions (Table 3).

The metabolisable energy content of the rotation depended statistically significantly on the rates of nitrogen, phosphorus, and potassium fertilisers (N (a1), P (a2), K (a3), N^2^ (a4), P^2^ (a5), K^2^ (a6)), and a significant relation was not obtained between NP (a7), NK (a8) and PK (a9) interaction.

### 3.2. Potassium Accumulation and Balance in Agricultural Crop Yields

Potassium accumulation in the yields of agricultural crops depended on the rates of potassium fertiliser, as well as on their fertilisation with nitrogen and phosphorus (Table 4). On average, annual fertilisation of agricultural crops with 96 and 192 kg ha^−1^ K_2_O, but without nitrogen and phosphorus fertilisation, resulted in an increase in potassium accumulation in the yield by 37.5 and 51.8%, respectively, compared to the potassium-unfertilised plots. In contrast, the average annual fertilisation of agricultural crops with all the major plant nutrients (N_108_P_96_K_96_) increased potassium accumulation in the crop yield by 121.4%.

The highest potassium accumulation in agricultural crops was obtained when fertilised with the maximum rates of nitrogen, phosphorus, and potassium fertilisers (N_216_P_192_K_192_). This resulted in a 214.3% increase in potassium accumulation in the crop yield compared to plots not fertilised with mineral fertilisers.

Potassium accumulation in crop yields was strongly and significantly (R = 0.96; *p* ≤ 0.05) influenced by nitrogen, phosphorus, and potassium fertiliser rates, and also by nitrogen and potassium interaction (Table 5). Nitrogen fertiliser had the greatest effect on potassium accumulation in the yield of agricultural crops, followed by potassium fertiliser to a lesser extent, and phosphorus fertiliser to the least extent.

Without potassium fertiliser, regardless of whether the plants were fertilised with nitrogen and phosphorus fertilisers or not, the annual balance of this element was negative, ranging from 56 to 90 kg ha^−1^ (Table 4). In contrast, a positive potassium balance was obtained when the plants were fertilised with a potassium fertiliser rate of 96 kg ha^−1^ but without nitrogen and phosphorus fertilisers. When the crops were fertilised with the above-mentioned rate of potassium and 108 kg ha^−1^ of nitrogen fertiliser, a negative potassium balance was obtained, due to the higher yields produced. The effect of phosphorus fertiliser on the potassium balance was less significant compared to nitrogen. When agricultural crops were fertilised with 216 kg ha^−1^ of nitrogen and 192 kg ha^−1^ of potassium, and when phosphorus was not used or applied at rates of 96 and 192 kg ha^−1^, the amounts of potassium incorporated into the soil were 54, 40, and 16 kg ha^−1^ more than the amounts of potassium accumulated by the crop yield, respectively. Annual fertilisation of agricultural crops with high rates of 192 kg ha^−1^ K_2_O resulted in a positive potassium balance in the soil; however, the combined application of nitrogen and phosphorus fertilisers resulted in a less significant potassium balance.

### 3.3. Uptake of Potassium from Fertilisers

The uptake of potassium from mineral fertiliser depended on the application of potassium and nitrogen, and to a lesser extent on the application of phosphorus (Table 6). When potassium was applied to agricultural crops at fertiliser rates of 96 and 192 kg ha^−1^ without nitrogen and phosphorus, the uptakes of potassium from the fertiliser were only 22.1 and 15.2%, respectively.

In contrast, the best uptake of potassium fertiliser by the plants (51.6%) was obtained when fertilised at N_108_P_96_K_96_. Increasing the average annual rate of potassium fertiliser to 192 kg ha^−1^ reduced its uptake, but not uniformly, and more potassium was taken up when the crops had not been fertilised with nitrogen and phosphorus fertilisers, or when fertilised at lower rates. Potassium uptake by agricultural crops at high rates of nitrogen and phosphorus (N_216_P_192_) was almost the same at the 96 and 192 kg ha^−1^ K_2_O fertiliser rates, i.e., 46.3 and 45.8%, respectively.

### 3.4. Accumulation of Potassium in Soil

The plant available potassium (K_2_O) content in the soil, as significantly determined by the Egner–Riehm–Domingo method and used as a reference in Lithuania, varied over a wide range of 183–630 kg ha^−1^ in the plots after 50 years of experimentation, and the maximum difference accounted for 3.4 times (Table 7). In the plots not fertilised with potassium fertiliser in 2020, the plant available K_2_O content was about 105–138 kg ha^−1^ lower than in 1971. The increase in plant available K_2_O was limited to the soil of plots where the plants had been under-fertilised with nitrogen and phosphorus fertilisers and where they had received at least 96 kg ha^−1^ of potassium each year.

When fertilising the agricultural crops with nitrogen fertiliser at rates of 72 kg ha^−1^ and above on average annually, along with phosphorus fertiliser, the plant available K_2_O content in the soil decreased in most of the treatment plots, except for those where the crops had been fertilised with potassium fertiliser at rates of 128 kg ha^−1^ and above.

After 50 years of experimentation, we also conducted tests on the 0–20 cm soil layer to assess soil non-exchangeable potassium using a more potent extractant, 1N HNO_3_ (Table 6). This method allowed us to detect potassium not only in the exchangeable form, but also in clay particles. The potassium content in that extract varied between 8268 and 9774 kg ha^−1^ in the experimental plots, which was 13–45 times higher than plant available K_2_O measured by the A–L method. The maximum difference in the content of soil non-exchangeable potassium found, depending on NPK fertilisation, was only 1890 kg ha^−1^ or 18% of the maximum content. The increase in non-exchangeable potassium found in the soil was also influenced by higher rates of potassium fertiliser but the influence of nitrogen and phosphorus fertilisers was less regular.

The regression analysis of the data revealed a very strong correlation (R = 0.93; *p* < 0.05) between the plant available K_2_O content in the soil determined by the A–L method and the nitrogen, phosphorus, and potassium fertiliser rates and no relation on their ratios (Table 8). The multinomial regression parameters show that the application of potassium fertilisers resulted in a significant increase of plant available K_2_O content in the soil, as determined by the A–L method, while the application of nitrogen and phosphorus fertilisers led to a decrease.

The correlation analysis showed a medium-tight, but statistically significant relationship (R = 0.68; *p* ≤ 0.05) between the content of soil non-exchangeable potassium and the rates of nitrogen, phosphorus, and potassium fertilisers.

## 4. Discussion

In the long-term (50-year) study, the application of potassium fertiliser was most effective on soils with low plant-available K_2_O content when used in combination with the other main plant nutrients—nitrogen and phosphorus. The average annual crop metabolisable energy (GJ ha^−1^) within the crop rotation was strongly (R = 0.97 *) dependent on the NPK fertiliser rates and their interactions. According to other authors, the interaction of nitrogen, phosphorus, and potassium fertilisers improved crop growth and yield, while the effectiveness of the interactions between nutrient elements depended on their relative content in soil [12,14]. Studies in three long-term fertilisation experiments on agricultural crops in Rothamsted, Bad Lauchstädt, and Skierniewice also showed that the effectiveness of potassium fertilisers decreased when plants experienced deficiencies in nitrogen, phosphorus, and magnesium [42]. Studies in China showed that the most effective rate of potassium fertiliser applied to agricultural crops was 120 kg ha^−1^, which increased maize and winter wheat yields by 16.7 and 25.1%, respectively [32]. It is also noted that potassium, like phosphorus, is very important for nitrogen uptake by agricultural plants. The balanced application of these fertilisers to agricultural crops resulted in a sixfold increase in root mass compared to plants solely fertilised with nitrogen fertiliser [12].

In our study, the fertilisation of agricultural crops with nitrogen fertiliser (N_108_P_0_K_0_) alone resulted in a 24.3% increase in metabolisable energy content, and fertilisation with all three plant nutrients (N_108_P_96_K_96_) resulted in a 98.8% increase in metabolisable energy content compared to unfertilised plots. Balanced fertilisation of agricultural crops resulted in higher crop yields, which in turn resulted in higher potassium accumulation. However, the high amounts of potassium that are removed from the soil pose a problem in the production of agricultural crops [43]. As Smill [18] points out, only 35% of potassium exported with cereal crop yields globally is compensated for with fertilisers, while other authors [44] have reported that only 10% of the potassium balance in the soil is compensated for. The negative soil potassium balance has been identified as a serious problem for food production at both the regional and global levels [45]. To address this problem partially, cereal straw should be left and ploughed in more often and potassium fertilisers should be used more efficiently, according to the recommendations that have been developed [8,46]. According to our research, when straw and other agricultural crop by-products are removed from the field, an average of about 160 kg ha^−1^ of potassium (K_2_O) should be applied annually to stabilise the plant-available K_2_O content in the soil.

The efficiency of potassium fertiliser use is reflected in its uptake.

It has been estimated that from 1961 to 2015, only 19% of potassium from fertiliser was taken up by cereals and food crops [8]. However, in our study, potassium uptake from fertiliser was significantly higher, at 51.6%, when agricultural crops were fertilised using a balanced fertiliser combination of 216 kg ha^−1^ of nitrogen and 96 kg ha^−1^ each of P_2_O_5_ and K_2_O. This indicates that there is still significant room for improvement in the efficiency of potassium fertiliser use worldwide. Moreover, similar results on potassium fertiliser uptake have been obtained by researchers in other countries. Long-term fertilisation experiments carried out in Rothamsted (UK), Bad Lauchstädt (Germany), and Skierniewice (Poland) have estimated potassium uptake from fertilisers to be between 44% and 62%, depending on the cation exchange capacity and the amount of clay in soil [42]. In the Czech Republic, two long-term fertilisation experiments on agricultural crops showed that the uptake of potassium from mineral fertilisers ranged from 27% to 52% [47].

Our study showed that an increase in the average rate of potassium fertiliser from 96 to 192 kg ha^−1^ K_2_O resulted in a decrease in potassium uptake of 51.6% to 45.8%. Similar patterns were found in a long-term (8-year) fertilisation experiment in China, where annual application of 48 kg ha^−1^ of potassium resulted in 39.5% uptake by plants. However, when the rate of potassium fertiliser application was increased to 156 kg ha^−1^, the uptake of potassium decreased to 16.4% [32]. The efficiency of potassium fertiliser use can be improved by taking into account the plant-available K_2_O content in the soil when fertilising, as studies in the USA have shown that it is not economically viable to fertilise agricultural crops with potassium fertilisers if the plant available K_2_O content in the soil is above the average, as they already have sufficient soil reserves of this plant nutrient [48]. Fertiliser application rates, balance, and uptake of potassium from fertilisers influence the changes in the amount of this element in the soil. Plots not fertilised with potassium, but fertilised with nitrogen and phosphorus, show a greater decrease in plant-available K_2_O compared to unfertilised plots. These trends have been observed in our long-term experiment, and studies in France have shown that in unfertilised plots, the decrease in plant-available K_2_O in the soil over a 25-year period is small and is due to the release of potassium from other forms. However, that decrease was lower than indicated by the potassium balance calculation [49]. Other researchers in long-term fertilisation experiments found that plant-available K_2_O levels in unfertilised plots decreased over 21 study years, but, perhaps due to potassium being released from the non-exchangeable form, these changes did not always accurately reflect the balance of potassium in soil [47]. In a long-term crop-fertilisation experiment in Rothamsted, UK, changes in soil potassium levels did not correlate with the balance of this plant nutrient [50]. Similar results were obtained in our study; an increase in plant-available K_2_O in soil in plots also did not always match the balance of this element. In addition, as a result of long-term fertilisation, there is an increase not only in the mobile but also in the non-exchangeable potassium content in the soil. Potassium in this form is in dynamic equilibrium with plant available potassium and potassium in the soil solution and is therefore also important in plant nutrition [51]. Our long-term studies have shown that when fertilizing with potassium fertilisers, it is necessary to optimise plant nutrition with phosphorus and potassium. This improves the efficiency of using potassium fertilisers and reduces the accumulation of potassium in the soil.

## 5. Conclusions

In the long-term (50-year) mineral NPK fertilisation experiment, the yield of agricultural crops, potassium balance, potassium use efficiency, and potassium content in soil depended not only on potassium fertilisation, but also on its interaction with nitrogen and phosphorus. The balance of potassium in soil was positive when agricultural crops had been fertilised with potassium fertilisers only. However, when nitrogen and phosphorus fertilisers were applied together, to compensate for potassium removal, when applying low nitrogen (72 kg ha^−1^) and phosphorus (64 kg ha^−1^) fertiliser rates, 128 kg ha^−1^ of potassium fertilisers is required. When using high nitrogen (180 kg ha^−1^) and phosphorus (160 kg ha^−1^) fertiliser rates, 160 kg ha^−1^ of potassium is needed. The highest potassium uptake, reaching 51.6%, was achieved when plants had been fertilised with nitrogen (108 kg ha^−1^), phosphorus (96 kg ha^−1^), and potassium (96 kg ha^−1^). When fertilising with potassium fertilisers alone, the content of plant-available K_2_O content in the soil increased, while with fertilisation with nitrogen and phosphorus combined, plant-available K_2_O is decreased, except in the plots where the plants had been fertilised with potassium fertiliser at rates of 128 kg ha^−1^ and higher. Due to the influence of fertilisers, the amount of non-exchangeable potassium in the soil also increased, but relatively little compared to the amount of available potassium. Thus, one of the main conditions for the effective use of potassium fertilisers is ensuring optimal plant nutrition with other nutrition elements, especially nitrogen and phosphorus.

## Figures and Tables

**Table 1 plants-12-03700-t001:** Soil properties of the experimental field in the 0–20 cm layer.

Study Indicators	Values ± SD
Clay particles (<0.002 mm) %	14.1 ± 0.65
Silt (0.002–0.05 mm) %	31.2 ± 2.17
Sand (0.05–2 mm) %	54.7 ± 2.72
Bulk density g cm^2^	1.50 ± 0.16
C_org_ %	1.58 ± 0.14
Effective exchange cation capacity me kg^−1^	130 ± 4.60
pH_KCI_	7.2 ± 0.56
Available P_2_O_5_ kg ha^−1^	171 ± 5.50
Available K_2_O kg ha^−1^	327 ± 11.1
Total K_2_O %	2.73 ± 0.10

Data prior to the test installation in 1971.

**Table 2 plants-12-03700-t002:** Effect of mineral fertiliser rates and their combinations on the productivity of agricultural crops in terms of average annual metabolisable energy.

Average Annual Fertilisation Rate kg ha^−1^								
N	P_2_O_5_	K_2_O						
		0	32	64	96	128	160	192
		Average annual metabolisable energy GJ ha^−1^						
0	0	40.3 a *			46.9 b			47.3 b
	96	49.1 bc			53.6 cdef			54.8 def
	192	51.5 bcde			53.6 cdef			55.3 defg
36	32		61.8 hijk				65.7 jkl	
	160		66.6 klmn				68.7 lmno	
72	64			72.2 nop		75.8 pqr		
	128			74.1 opq		77.6 pqrs		
108	0	50.1 bcd			58.3 fghi			64.4 jkl
	96	63.2 ijkl			80.1 rstu			82.8 stuvw
	192	65.6 jkl			82.6 stuvw			84.1 tuvw
144	64			78.2 qrst		81.6 rstuvw		
	128			80.4 rstu		85.5 uvw		
180	32		71.8 mnop				77.2 pqrs	
	160		76.4 pqr				85.4 uvw	
216	0	57.0 efgh			64.1 jkl			60.6 ghij
	96	65.8 jkl			82.5 stuvw			85.2 uvw
	192	65.9 jklm			83.1 stuvw			87.3 w

*—significant differences were established between the data lettered a, b, c, d, e, f, etc., at 5% probability level (*p* ≤ 0.05).

**Table 3 plants-12-03700-t003:** Dependence of the metabolic energy (GJ ha^−1^) on NPK fertiliser rates (kg ha^−1^) and their interactions.

Equation y = a_0_ + a_1_N + a_2_P + a_3_K + a_4_N^2^ + a_5_P^2^ + a_6_K^2^ + a_7_NP + a_8_NK + a_9_PK parameters										R
a_0_	a_1_	a_2_	a_3_	a_4_	a_5_	a_6_	a_7_	a_8_	a_9_	
39.8	0.24 *	0.19 *	0.13 *	−0.00085 *	−0.00081 *	−0.00059 *	0.00022	0.00023	0.00016	0.94 *
*p*										
	<a1	<a2	<a3	<a4	<a5	<a6	0.0020	0.0016	0.046	≤0.05

*—correlation statistically significant at the 95% probability level (*p* ≤ 0.05).

**Table 4 plants-12-03700-t004:** Effect of long-term fertilisation on potassium accumulation in crop yields and its balance in the soil.

Average Annual Fertilisation Rate kg ha^−1^								
N	P_2_O_5_	K_2_O						
		0	32	64	96	128	160	192
		Potassium (K_2_O) accumulation in agricultural crops yields kg ha^−1^/balance kg ha^−1^						
0	0	56 a */−56			77 cd/19			85 de/107
	96	63 ab/−63			86 de/10			103 ghi/89
	192	72 bc/−72			90 ef/6			100 fg/92
36	32		86 de/−54				110 ghij/50	
	160		102 gh/−70				118 jkl/42	
72	64			104 ghi/−40		125 lmn/3		
	128			114 ijkl/−50		143 op/−15		
108	0	78 cd/−78			113 hijk/−17			132 mno/60
	96	83 de/−83			124 klmn/−28			158 qr/34
	192	90 ef/−90			133 mno/−37			165 rst/27
144	64			122 klm/−58		150 pq/−22		
	128			125 lmn/−61		153 pq/−25		
180	32		113 hijk/−81				135 no/25	
	160		116 jkl/−84				161 qr/−1	
216	0	84 de/−84			119 jkl/−23			138 o/54
	96	84 de/−84			133 mno/−37			152 pq/40
	192	89 ef/−89			133 mno/−37			176 t/16

*—significant differences were established between the data lettered a, b, c, d, e, f, etc., at 5% probability level (*p* ≤ 0.05).

**Table 5 plants-12-03700-t005:** Dependence of potassium (K_2_O) accumulation (kg ha^−1^) in agricultural crop yields on NPK fertiliser rates (kg ha^−1^) and their interaction.

Equations y = a_0_ + a_1_N + a_2_P + a_3_K + a_4_N^2^ + a_5_P^2^ + a_6_K^2^ + a_7_NP + a_8_NK + a_9_PK * parameters										R
a_0_	a_1_	a_2_	a_3_	a_4_	a_5_	a_6_	a_7_	a_8_	a_9_	
55.0	0.46 *	0.14 *	0.31 *	−0.0017 *	−0.0005	−0.0009 *	0.0001	0.0009 *	0.0005	0.96 *
*p*										
	<a1	<a2	<a3	<a4	0.018	<a6	0.494	<a8	0.0017	≤0.05

*—correlation statistically significant at the 95% probability level (*p* ≤ 0.05).

**Table 6 plants-12-03700-t006:** Effect of mineral fertiliser rates and their combinations on the uptake of potassium.

Average Annual Fertiliser Rate kg ha^−1^				
N	P_2_O_5_	K_2_O		
		0	96	192
		Uptake of potassium from fertiliser %		
0	0	–	22.1	15.2
	96	–	24.2	21.0
	192	–	18.9	14.7
108	0	–	36.8	28.4
	96	–	43.1	39.6
	192	–	45.3	39.5
216	0	–	36.8	28.4
	96	–	51.6	35.8
	192	–	46.3	45.8

**Table 7 plants-12-03700-t007:** Effect of mineral fertiliser rates and combinations on soil potassium levels.

Average Annual Fertilisation Rate kg ha^−1^								
N	P_2_O_5_	K_2_O						
		0	32	64	96	128	160	192
		Soil potassium A–L/non-exchangeable kg ha^−1^						
0	0	222 cdef */ 8583 abc			426 nop/ 9699 def			576 t/ 9774 f
	96	231 defg/ 8889 abcdef			363 kl/ 9327 bcdef			630 u/ 9195 abcdef
	192	198 abc/ 8277 ab			402 mn/ 9318 bcdef			567 st/ 9333 cdef
36	32		264 h/–				561 st/–	
	160		246 efgh/–				549 st/–	
72	64			267 h/–		420 no/–		
	128			261 gh/–		474 qr/–		
108	0	219 bcde/ 9621 cdef			402 mn/ 8943 abcdef			456 pq/ 9534 cdef
	96	192 ab/ 8268 a			366 kl/ 9354 cdef			450 opq/ 9675 def
	192	183 a/ 9207 abcdef			330 ij/ 9255 abcdef			573 st/ 9507 cdef
144	64			252 fgh/–		432 nop/–		
	128			225 cdef/–		366 kl/–		
180	32		240 efgh/–				420 no/–	
	160		192 ab/–				492 r/–	
216	0	189 a/ 8916 abcdef			384 lm/ 9306 bcdef			543 s/ 9627 cdef
	96	183 a/ 8667 abcdef			342 jk/ 9033 abcdef			501 r/ 9234 abcdef
	192	204 abcd/ 9066 abcdef			306 i/ 8604 abcd			426 nop/ 9294 bcdef

Before the experiment was set up in 1971, the average amount of available K_2_O in the soil was 327 ± kg ha^−1^; *—significant differences were established between the data lettered a, b, c, d, e, f, etc., at 5% probability level (*p* ≤ 0.05).

**Table 8 plants-12-03700-t008:** Dependence of soil potassium content (kg ha^−1^) on NPK fertilization rates and their interaction.

Equation y = a_0_ + a_1_N + a_2_P + a_3_K + a_4_N^2^ + a_5_P^2^ + a_6_K^2^ + a_7_NP + a_8_NK + a_9_PK * parameters										R
a_0_	a_1_	a_2_	a_3_	a_4_	a_5_	a_6_	a_7_	a_8_	a_9_	
Plant available K_2_O										
227.8	−0.49 *	−0.29 *	1.96 *	0.002 *	0.001 *	−0.00012	−0.0008	−0.002	0.0007	0.93 *
*p*										
	<a1	<a2	<a3	<a4	<a5	0.351	0.912	0.252	0.0051	<0.05
Non-exchangeable K_2_O										
8901	2.95 *	−4.24 *	5.56 *	−0.012	0.015	−0.0061	0.0019	−0.0084	−0.0021	0.68 *
*p*										
	<a1	<a2	<a3	0.323	0.324	0.695	0.841	0.394	0.849	<0.05

*—correlation statistically significant at the 95% probability level (*p* ≤ 0.05).

## Data Availability

All data are included in the present study.

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
