# Peer review of "The Influence of Mineral NPK Fertiliser Rates on Potassium Dynamics in Soil: Data from a Long-Term Agricultural Plant Fertilisation Experiment"

_plants, 2023, doi:10.3390/plants12213700_

Round 1

Reviewer 1 Report

Dear authors, the manuscript entitled "The influence of mineral NPK fertilizer rates and their ratio on potassium dynamics in soil: data from a long-term agricultural plant fertilization experiment" presents interesting results, based on the data extracted from a long-term field of observations.

There are some changes and suggestions that can improve the manuscript.

Introduction - The last paragraph should be rewritten to present a clear aim and the hypotheses proposed and tested by the authors. This is necessary for a better understanding of the manuscript. Make the sentences shorter in the manuscript 2-3 rows, to be focused on separate concepts.

Results section - this section is interesting. I suggest the authors to make smaller tables, based on the format of the journal.

Discussion section is well balanced and connect the results with international literature.

Material and Methods describe in detail the experimental setup and the methods used by the authors.

Conclusion section provide a good synthesis of results and the most important findings, based on collected data.

Overall, the article is balances and well written.

Author Response

Thank you very much for taking the time to review this manuscript. Please see the attachment.

Reviewer 2 Report

In this paper, the effect of different potassium fertiliser rates and their interaction with nitrogen and phosphorus of K by various crops is discussed with the data of 50 years of agricultural fertilization experiment. The research results provide an important reference for field rotation, potassium balance and fertiliser utilization. 

Some contents in this paper need to be improved, and some results need further analysis and discussion. The comments and suggestions for revision of each part are as follows.

Title: It is suggested to delete “and their ratio” from the title. “The NPK fertilizer ratio” is included in “the NPK fertilizer rates”.

Abstract: There is no obvious conclusion at the end of the abstract. For example, how to apply potassium fertilizer or NPK interaction mechanism.

Introduction: Add the relevant references on the impact of lack or excessive application of potassium fertilizer on agricultural yield.

Line 94-98  Incorrect typesetting format

Results: It is suggested that original data of 50 years should be provided in supplementary data.

In 50-year research, how to eliminate the influence of climate factors on the experimental results?

It is suggested that all tables be changed to three-line tables.

The results of root length determination of potted plants are not accurate, so it is recommended to use root fresh weight. The number of sample replicates is not indicated below the table.

Line 139: The meaning of this sentence “ significant relation not obtained between those nutrients” is not clear.

Discussion: It is suggested to add a conclusion at the end of the discussion to summarize the innovation and significance of this study.

Author Response

(The authors gave the same response as above.)

Reviewer 3 Report

In this research manuscript, the authors conducted a meticulous fertilization experiment to investigate the effects of different potassium fertilizer rates. With a particular focus on field rotation productivity, potassium balance, fertilizer utilization, and changes in soil potassium content, the experiment aimed to provide valuable insights into crop production.

However, while the study successfully presented its findings through tables, there are a few areas where further clarification is needed. In lines 94-98, an inconsistency in the font size of certain words is observed, and the sentence in question remains incomplete. It would be beneficial for the authors to address this matter to ensure the manuscript’s coherence and readability.

Another aspect that requires attention is the absence of comprehensive explanations for some of the presented parameters. For instance, the metabolizable energy parameter lacks clarification regarding its calculation, meaning, and implications within the context of the study. By providing a detailed explanation of such parameters, the authors can enhance the understanding and interpretation of their findings.

Furthermore, in table 2, additional information is required to elucidate the specific meanings and significance of all the parameters presented. This additional information would greatly contribute to the readers’ comprehension of the experimental results and their implications for crop production.

Therefore, to further improve the manuscript’s quality and effectiveness, it is recommended that the authors address these concerns and provide the necessary explanations and information to ensure a comprehensive understanding of the research findings and their implications.

Author Response

(The authors gave the same response as above.)

Reviewer 4 Report

Dear Authors, manuscript entitled „The influence of mineral NPK fertilizer rates and their ratio on  potassium dynamics in soil: data from a long-term agricultural  plant fertilization experiment” was submitted to journal “Plants” Section of “Horticultural Science and Ornamental Plants” what in my opinion is an obvious mistake because the manuscript is focused only on soil and agricultural not horticultural plants (plants’ tissues were analysed for potassium content and  data were presented as a total sum of metabolic energy and potassium accumulation therefore my suggestion is to submit this manuscript to agronomic journals. In my opinion the next obvious mistake is structure of the manuscript because section Materials and Methods should be presented just after Introduction but not as the next after Discussion section.

Coming to the content of the manuscript it has to be mentioned that it presents results from long-lasting field experiments and therefore values of results are very high however in my opinion using only two replications can limit this values of some extent. Applied methods are standard ones however method used for determining extractable potassium was Egner-Riem – Domingo while Olsen’s method is more widely used. Reference no 49 was wrongly used in line 389 because the Reference concerns soil not plant analysis.

Sections Results and Discussion are strong points of the manuscript. Analytical results are presented in tables only. Results are presented in a clear way but tables design should be improved.

Reviewer as non-native English speaker is not competent to evaluate English language level but I would like to suggest language polishing by native English speaker.

Some detailed remarks:

Reference nr 29 (Phosphate rock. Mineral commodity sumamries, January 2016) was irrelevant in the context of paragraph – line 77

Lines 94-98 -was written in a different character type than the rest of the document.

Sentences in lines 262-264; 286-287; 327-330 are unclear.

Author Response

(The authors gave the same response as above.)

Round 2

Reviewer 3 Report

The author have revised the manuscript.